# Factors predicting students' performance in the final pediatrics OSCE

**Maysoun Al Rushood****\*, Amal Al-Eisa**

Department of Pediatrics, Faculty of Medicine, Health Sciences Center, Kuwait University, Kuwait City, Kuwait

\* maysounrushood@hsc.edu.kw

## Abstract

### Background

Objective Structured Clinical Examinations (OSCEs) have been used to assess the clinical competence of medical students for decades. Limited data are available on the factors that predict students' performance on the OSCEs. The aim of our study was to evaluate the factors predicting performance on the pediatrics final OSCE, including the timing of students' clerkship and their performance on the in-training OSCE and written examinations.

### Methods

Grades in pediatrics for 3 consecutive academic years (2013–2016) were included. The average scores of the in-training OSCEs, written and final OSCEs and written exams were compared among the three years using the analysis of variance (ANOVA) test. The correlations between performance on the final OSCEs and the in-training OSCEs, in-training written exams and final written exams were studied using Spearman's Rho correlation test. The effect of the timing of the clerkship on the final OSCE performance was evaluated.

### Results

A total of 286 students' records were included. There were 115 male students and 171 female students (M:F 1:1.5). There were strong positive correlations between students' performance on the in-training examinations (OSCE and written) and the final OSCE (correlation coefficients of 0.508 and 0.473, respectively). The final written exam scores were positively correlated with the final OSCEs (r = 0.448). There was no significant effect of the timing of the clerkship.

### Conclusions

Students' performance on in-training examinations might predict their final OSCE scores. Thus, it is important to provide students with the necessary intervention at an early stage to reduce failure rates. The final OSCE performance does not seem to be affected by the timing of the clerkship.

**Data Availability Statement:** All relevant data are within the manuscript and its Supporting Information files.

**Funding:** the authors receive no specific funding for this work.

**Competing interests:** the authors have declared that no competing interests exist.

## Introduction

OSCE stands for Objective Structured Clinical Examination. OSCEs have been used to assess the clinical competence of health care professional students for decades [1, 2]. They involve a series of encounters objectively testing clinical skills including history taking, physical examination, communication and data interpretation [3]. The OSCE is a feasible approach to assess clinical competency due to its inherent flexibility [4, 5]. The reliability and utility of this examination have been widely studied [4–11].

Studying the factors related to students that affect their performance is highly important in order to provide the learners with the optimal educational conditions for the best outcomes in terms of knowledge retention, application and exam performance. The correlations between OSCE performance and measures such as performance on future OSCEs, USMLEs and medical school grade point average have been reported [6–9]. However, limited data are available on some other important student-related factors affecting their OSCE performance, such as the effects of the timing of one's clerkship during the academic year, the performance on the written examination components and the performance on the in-training OSCEs [1, 3, 12, 13]. Such correlations are crucial to evaluate if the students have improved their performance throughout the academic year. We believe that students should demonstrate improved skills with each OSCE undertaken. This has important implications for medical schools in providing students with poor in-training performance with the necessary intervention at an early stage to reduce failure rates and improve students' outcomes.

The aim of our study is to evaluate the factors affecting students' performance on the Pediatrics Final Year OSCE, including the timing of their clerkship, their performance on the in-training OSCE and their written examination score.

## Materials and methods

This descriptive study was conducted with 6[th] year clinical students at the Faculty of Medicine, Kuwait University, and it analyzed the grades in the pediatrics rotations for 3 consecutive classes of students for three academic years (2013–2014, 2014–2015, and 2015–2016).

The School of Medicine, Kuwait University, admits an average of 100 students every year and offers a 7-year undergraduate teaching program. The faculty curricula, for both preclinical and clinical programs, were reformed in 2005–2006, and consequently, problem-based learning was introduced [14].

### Study setting

In their 6th year, students are divided into 3 groups. They rotate between pediatrics, obstetrics and gynecology and 4 other subspecialties in medicine and surgery. There is one pediatric rotation completed by the students. Sixth year medical students spend 12 weeks in pediatrics. Students complete clinical rotations in the general pediatric wards along with 2 weeks in the NICU in the 4 main teaching hospitals in the country; each student rotates in 2 hospitals. Clinical teaching comprises bedside teaching, procedure and communication skills teaching. In addition, the students join the medical team in the ward, where they are assigned patients to follow and present during ward rounds.

In addition to the clinical teaching, there are seminars, medical school days and paper-based problem-based learning (PBL) sessions delivered at the faculty campus.

At the end of the pediatrics rotation, there is an end-of-block examination with both written (computer-based MCQs) and OSCE components. The final exam, which is carried out at the end of the year, is composed of similar components. Six external examiners from Europe

and North America are invited every year to assess our students on the final OSCE. External examiners are invited only for the final exam every year.

The OSCE is performed on real pediatric patients, healthy children and simulated parents in the hospital. We run 2 successive cycles with identical stations. The end-of-block OSCE is run in 2 teaching hospitals in one day, while the final OSCE is conducted over 2 days in the 4 main teaching hospitals each day. To maintain consistency, examiners are given protected time immediately before the exam to agree on the physical signs and what the student must demonstrate in order to achieve the score, thereby standardizing the marking process.

The exam consists of 9 clinical stations: neurology, pulmonology, gastroenterology, cardiology, communication skills, history taking, clinical procedure demonstration, normal development assessment and one other station that could be genetics or dermatology. Except for the history taking, which is a 20 minute station, the other stations are 10 minutes in duration. One to two minutes are allowed between stations for students to move and examiners to complete their marks. In addition to a checklist, a global mark is awarded based on fluency and mastery. The department of pediatrics holds an orientation session for the external examiners to explain the guidelines of the exam and what standards are expected from the students. The grading sheets and the global assessment sheets are discussed. Examiners are briefed on each day of the OSCE on the standards expected. The pass mark is set before the exam and no changes on the marks are made after the exam. The grades are expressed as A ($\geq$ 90%), B (80–89%), C (60–79%) and F ($<$60%). All grades are expressed as percentages. The in-training OSCE is graded, and feedback is provided to failing students at a later date; however, students complete Mini-Cex sessions during their training, which are not graded but are given immediate feedback.

Ethical approval was obtained from the Health Sciences Center Ethical Committee, Faculty of Medicine, Kuwait University. The Board members are the Vice Dean of Research at the Faculty of Medicine, who is the head of the committee, and five other members who are full professors in different specialties. The use of students' grades was approved by the Vice-Dean Academic Office at the Faculty of Medicine. The confidentiality of the identity of the students included was ensured.

## Statistical analysis

The data management, analysis and graphical presentation were carried out using the computer software 'Statistical Package for Social Sciences, SPSS version 25.0' (IBM Corp, Armonk, NY, USA). The descriptive statistics for students' overall academic grade performance during the three years were presented as numbers and percentages. The average scores were computed and presented as the means ± standard deviations (SD) with a range for each examination component, and the three years were compared using analysis of variance (ANOVA) with the Bonferroni test for multiple comparisons. The Shapiro-Wilk test was used to check for a normal distribution for all the data studied. Spearman's rho correlation test was used to find the correlations of the final OSCE performance and the in-training OSCE, in-training written and final written performances. The one-way ANOVA test was used to evaluate the effect of the order of the clerkship on the final OSCE performance. A two-tailed probability value 'p' $<$ 0.05 was considered statistically significant.

## Results

A total of 286 students' score records were included in the study. The distribution of the number of the students across three consecutive years is presented in Table 1 (The total number of male students was 115 and the number of female students was 171 (M:F ratio 1:1.5)).

**Table 1. Mean score for the end of block, final OSCE, end of block written and final written by year.**

| Year of examination | Number of students | Gender Male | Gender Female | End of Block OSCE | End of block written | Final OSCE | Final written |
|---|---|---|---|---|---|---|---|
| | | No (%) | No (%) | Mean±SD | Mean±SD | Mean±SD | Mean±SD |
| 2013–14 | 92 | 37 (40.2) | 55 (59.8) | 77.96±7.2 | 67.6±7.6 | 77.71±6.2 | 74.3±6.5 |
| 2014–15 | 90 | 30 (33.3) | 60 (66.7) | 80.7±5.5 | 69.3±8.1 | 78.4±6.3 | 74.5±7.5 |
| 2015–16 | 104 | 48 (46.2) | 56 (53.8) | 79.3±6.2 | 66.2±8.5 | 77.8±6.4 | 75.8±6.9 |
| Total | 286 | 115 (40.2) | 171(59.8) | 79.31±6.4 | 67.66±8.2 | 77.95±6.3 | 74.9±7.0 |

Students' grades in each exam component in each year are presented as the means ± SDs in Table 1. No significant difference was noticed in the final grades during the three years (P = 0.701).

The final grades for all the students in individual years and collectively are displayed in Fig 1. Overall, the most scored grade was C (68.9%) followed by B (25.5%).

## Performance on the OSCE (Table 2)

Over the whole sample, the results showed that there is a positive correlation between the performances on the end-of-block OSCE and the final OSCE with a correlation coefficient r of 0.508. This holds true for each academic year as well, where 2013–14, 2014–15, and 2015–16 had correlation coefficients of 0.539, 0.434, and 0.536, respectively (Table 2).

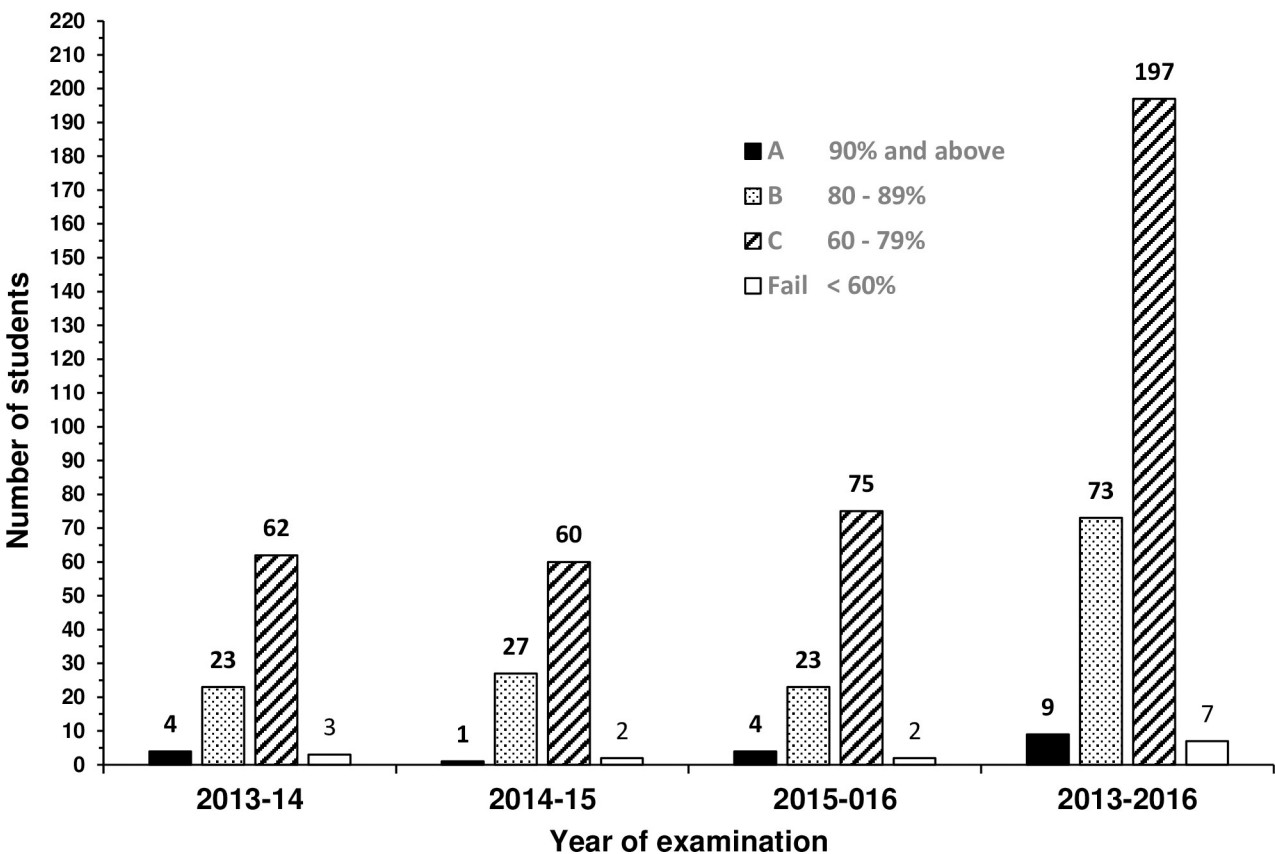

**Fig 1. Distribution of the final grades for the students in individual years and collectively.**

**Table 2. Spearman's rho correlations between the final OSCE and the end of block OSCE, final written and end of block written.**

| Academic year | Examination | Correlation Coefficient | P value |
|---|---|---|---|
| 2013–14 | End of Block OSCE | 0.539 | <0.001 |
| | Final written | 0.402 | <0.001 |
| | End of Block written | 0.428 | <0.001 |
| 2014–15 | End of Block OSCE | 0.434 | <0.001 |
| | Final written | 0.570 | <0.001 |
| | End of Block written | 0.574 | <0.001 |
| 2015–16 | End of Block OSCE | 0.536 | <0.001 |
| | Final written | 0.356 | <0.001 |
| | End of Block written | 0.428 | <0.001 |
| Total | End of Block OSCE | 0.508 | <0.001 |
| | Final written | 0.448 | <0.001 |
| | End of Block written | 0.473 | <0.001 |

## Performance on the written exam (Table 2)

The results showed a positive correlation (r = 0.448) between the overall scores on the written final and the final OSCE examinations (Table 2). A similar finding was noticed when the end-of-block written was compared to the final OSCE in each year and over the whole sample.

## Timing of clerkship (Table 3)

Looking at the timing of their clerkships, there was no significant effect on the performance on the final OSCE (Table 3).

The effect of the timing of the clerkship was studied for each academic year and the overall sample. When the means of the final OSCE performances for all the students in the 3 academic years (2013–2016) were compared, there was a statistically significant difference in the final OSCE performance for students who did their rotation last compared to those who did it in the middle of the year but not first (Table 3). The mean score was 79.17 in rotation 3 compared to 77.19 in rotation 2 with a P = 0.029.

## Discussion

Efforts are continuously being made to improve the assessment of medical students. The OSCE has been widely implemented due to its validity and reliability. It has added greater

**Table 3. Mean final OSCE scores by timing of clerkship.**

| Year of Examination | First | Second | Third | Total Rotation |
|---|---|---|---|---|
| | Mean ± SD | Mean ± SD | Mean ± SD | Mean ± SD |
| 2013–14 | 78.23±5.59 | 76.15±7.66 | 78.62±4.97 | 77.71±6.16 |
| | (n = 32) | (n = 29) | (n = 31) | (n = 92) |
| 2014–15 | 77.76±6.09 | 77.23±5.93 | 80.10±6.55 | 78.35±6.26 |
| | (n = 29) | (n = 31) | (n = 30) | (n = 90) |
| 2015–16 | 76.54±7.20 | 78.03±5.93 | 78.85±5.97 | 77.81±6.41 |
| | (n = 35) | (n = 34) | (n = 35) | (n = 104) |
| Total | 77.47±6.34 | 77.19±6.49 | 79.17±5.84* | 77.95±6.26 |
| | (n = 96) | (n = 94) | (n = 96) | (n = 286) |

* Statistically significant difference between third and second, P = 0.029.

objectivity to students' evaluation. Therefore, it is important to understand the factors that might affect students' performance and the ways to improve such performance to ensure the acquisition of the clinical skills and ability to apply medical knowledge.

In our study, we have demonstrated that there are strong correlations between students' performance on prior examinations in pediatrics rotations, both OSCE and written, and the final OSCE, as well as with the final written exam component.

Studies on correlations between formative and summative OSCEs have reported conflicting findings [1, 12–14]. Chisnall et al. found that formative OSCEs had a positive predictive value of 92.5% for passing the summative OSCE. Although they have shown that formative OSCEs were associated with improved performance only for identical stations in subsequent summative OSCEs, there was an improvement in the passing rates in the summative exam in other stations as well [14].

Previous reports have shown that repetitive testing enhances knowledge retention [12, 15]. This is extremely crucial in medical education, where the aim is to ensure the attainment of clinical skills for safe practice by the medical graduates. In fact, many previous studies supported the idea of assessment driving learning [16, 17].

Our findings of strong correlations of the final OSCE with the written exam component are consistent with previous studies [3, 8, 18–20]. This correlation emphasizes the importance of having a good knowledge base when taking the OSCE.

Couto et al. reported positive correlations between the summative OSCE and formative assessments, including assessments done during PBL sessions [19]. In addition, a study evaluating the predictors of OSCE performance, concluded that students who took a practice National Board of Medical Examiner (NBME) shelf exam did better on both their shelf exam and OSCE [20].

In our study, this correlation might be attributed to the fact that our OSCE is based on real pediatric patients with clinical findings; therefore, students learn to perform in a way that is relevant to the clinical findings. Clinical findings should be sought as the student progresses with patient encounters. They need to give explanations and answer a few questions at the end of the station. The ultimate objective of students' evaluation for a medical certificate is not only to evaluate the depth and breadth of their knowledge but more importantly to evaluate their ability to apply such knowledge and to correlate it to clinical findings. Our OSCE is clinical performance concentrated.

Previous studies have demonstrated that the closer the students' rotation is to their final exam, the better their scores were [1, 3, 8]. In our study, the timing of the clerkship did not affect the performance on the final OSCE when looking at individual years. However, when we compared the overall sample size, students who did their rotation at the end of the academic year had slightly better performance than those who did it in the middle of the academic year but not those students who completed their clerkship at the beginning of the year. In our study, this might be attributed to the fact that a clerkship in the middle of the year is interrupted by many holidays. Nonetheless, this was not true when comparing the third rotation to the first or the second to the first. We may conclude that there was no effect of the timing of the rotation on the final OSCE performance. Therefore, dividing students to complete different rotations throughout the year, according to the capacity of the training centers, staff availability, and applied curriculum, should not affect their performance on final exams. It seems that students have good long-term clinical capacities. This is extremely crucial in medicine, as our aim is for students to be able to retrieve the information when needed and to apply it later in their practice.

This is the first study done in the Faculty of Medicine at Kuwait University correlating in-training exam scores with final OSCE scores. Our data over the years were comparable. We were able to include the complete score records for entire batches of the period of study.

The limitations of the study include the lack of a comparison group. It would be imperative to study the outcomes of students' performance when there are no in-training assessments under the same educational settings. However, this was not feasible since there is only one school of medicine in the country. Comparing the grades between different specialties would not be conclusive. The OSCEs differed in their contents over the years. Although we kept similar components (systems to be examined), the exact cases were different. This might be a potential limitation, but it is designed this way to ensure that students do not know the exact content of the exam, which might affect their success in the examinations.

## Conclusion

In conclusion, there are positive correlations between students' performance on the final OSCE and their performance on both the in-training OSCE and the written examinations. Such correlations might be used as predictors of final OSCE grades. Use of these predictors would offer a better opportunity to provide students with the necessary intervention at an early stage to reduce failure rates. The positive correlation between the written exam components and the OSCE supports that adequate performance on OSCEs requires a good knowledge base. Final OSCE performance may not be affected by the timing of the clerkship.

## Supporting information

**S1 Dataset.**
(DOCX)

**S2 Dataset. Edited-final overall grading 13–14.**
(XLS)

**S3 Dataset. Edited-final overall grading 14–15.**
(XLS)

**S4 Dataset. Edited-final overall grading 15–16.**
(XLS)

## Acknowledgments

The authors would like to gratefully acknowledge Ms. Asiya Tasneem Ibrahim and Dr. Prem Sharma for their assistance with the statistical analysis. We would like to acknowledge Mr. Thomas M. De Souza for his great help in extracting the students' grades.

## Author Contributions

**Conceptualization:** Maysoun Al Rushood, Amal Al-Eisa.

**Data curation:** Maysoun Al Rushood.

**Formal analysis:** Maysoun Al Rushood, Amal Al-Eisa.

**Investigation:** Amal Al-Eisa.

**Methodology:** Maysoun Al Rushood, Amal Al-Eisa.

**Project administration:** Maysoun Al Rushood, Amal Al-Eisa.

**Supervision:** Maysoun Al Rushood, Amal Al-Eisa.

**Validation:** Maysoun Al Rushood.

**Visualization:** Maysoun Al Rushood, Amal Al-Eisa.

**Writing – original draft:** Maysoun Al Rushood.

**Writing – review & editing:** Maysoun Al Rushood, Amal Al-Eisa.

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
