## [Decision Letter · Decision Letter 0]

26 Jun 2020

PONE-D-19-27962

Factors Predicting Students’ Performance in the Final Pediatrics OSCE

PLOS ONE

Dear Dr. Al Rushood,

Thank you for submitting your manuscript to PLOS ONE. After careful consideration, we feel that it has merit but does not fully meet PLOS ONE’s publication criteria as it currently stands. Therefore, we invite you to submit a revised version of the manuscript that addresses the points raised during the review process.

We would appreciate receiving your revised manuscript by Jul 03 2020 11:59PM. To enhance the reproducibility of your results, we recommend that if applicable you deposit your laboratory protocols in protocols.io, where a protocol can be assigned its own identifier (DOI) such that it can be cited independently in the future. For instructions see: http://journals.plos.org/plosone/s/submission-guidelines#loc-laboratory-protocols

We look forward to receiving your revised manuscript.

Kind regards,

Oathokwa Nkomazana, MD MSC PhD

Academic Editor

PLOS ONE

Additional Editor Comments:

Thank you for an interesting article.

Please address the comments by the reviewer. In addition please address the following:

1. In the Methods section, it will be helpful to have a section titled: Study setting where you describe the setting of the medical school, where the clinical training is done, focusing on paediatrics; how the training is done; How many paediatric rotations do the students do in the course of training?Are the PBL sessions paper based or are they based on real patients? What is entailed in clinical teaching``/

How are the examination quality assured? You only talk to external examiners in the final OSCE. Is there blue printing of the examination? Is there moderation? Is it internal or are there aspects of external moderation? What does it mean that "Examiners are provided protected time just before exams?" what are the examiners briefed on?

In the data collection, make it clear if you followed the same cohort of students over three years, which is what I think you are reporting, but it's not clear. You say that you enrol 100 students a year but do not explain why you do not have 300 students in your study. Who have you excluded?

In the results section, you say that students who did their rotation last performed better than those who did it second, but not not better that those who did it first. Hw do you explain the difference? Could this be attributed to the differences in the assessments?

In the discussion,, the 7th paragraph that talks to differences in performance based on the timing of the rotation, the inferences are not very clear and it's not clear how this is supported by the reported findings. You conclude the paragraph by applauding the recall of information,; in medical education, however, recall is not considered a very high value but rather information literacy and life ling learning capacity as the half life of medical information is very short.

Journal Requirements:

3. Thank you for stating in the manuscript Methods section: 

'Ethical approval was obtained from The Health Sciences Ethical Committee. The use of students’ grades was approved by the Vice-Dean Academic Office at the Faculty of Medicine. Confidentiality of identity of students included was secured.'  

6. Please include your tables as part of your main manuscript and remove the individual files.

Please note that supplementary tables should be uploaded as separate "supporting information" files.

Reviewers' comments:

Reviewer's Responses to Questions

**Comments to the Author**

1. Is the manuscript technically sound, and do the data support the conclusions?

Reviewer #1: Partly

2. Has the statistical analysis been performed appropriately and rigorously? 

Reviewer #1: Yes

3. Have the authors made all data underlying the findings in their manuscript fully available?

Reviewer #1: No

4. Is the manuscript presented in an intelligible fashion and written in standard English?

Reviewer #1: Yes

5. Review Comments to the Author

Reviewer #1: I read the manuscript with great interest, I suggest the following comments to make it publishable

• There is need to address other factors that may affect the final OSCE like gender, day of the exam (the authors mentioned that they have around 30 students /group and there were 9 stations run in 2 cycles, this mean that the exam either repeated for three days or in three different place), studying these factors will add to the quality of paper.

• Was the in-training OSCE formative without marks, did you provide any feedback for student in the in-training OSCE

• The term error bar is mentioned in the method section but it is not shown in the figure.

• In result section (‘End of block OSCE’ and ‘End of block written’ showed some improvement, especially in the year 2014-15 compared to 2013-14 and 2015-16 (p=0.016 & p=0.031) respectively.) could you clarify this sentence more, what it means and refer to table number. The word improvement here is not perfect with the meaning. (improvement mean that student score improved in this test in comparison to previous test)

• In discussion section, the authors should shed light on the low r value between written and final OSCE, usually the OSCE should be a test of clinical competence and not knowledge try to discuss this point when comparing final OSCE and written test, could be the good correlation between written test and OSCE due to more knowledge concentrated OSCE stations rather than performance OSCE stations.

• To mention reliability of a test there is need for reliability index.

• In conclusion part the first sentence should be removed as it was not detected in the results.

• Try to use some new references that addressed similar problem.

Alkhateeb, N. E., Al-Dabbagh, A., Ibrahim, M., & Al-Tawil, N. G. (2019). Effect of a Formative Objective Structured Clinical Examination on the Clinical Performance of Undergraduate Medical Students in a Summative Examination: A Randomized Controlled Trial. Indian pediatrics, 56(9), 745–748.

English editing is advisable

6. PLOS authors have the option to publish the peer review history of their article (what does this mean?). If published, this will include your full peer review and any attached files.

Reviewer #1: No

---

## [Author Response · Author response to Decision Letter 0]

1 Jul 2020

1. In the Methods section, it will be helpful to have a section titled: Study setting where you describe the setting of the medical school, where the clinical training is done, focusing on paediatrics; how the training is done; How many paediatric rotations do the students do in the course of training?Are the PBL sessions paper based or are they based on real patients? What is entailed in clinical teaching``/

Our response: A section entitled “Study Setting” was added with the required details (lines 77-87 pages: 4,5).

2. How are the examination quality assured? You only talk to external examiners in the final OSCE. Is there blue printing of the examination? Is there moderation? Is it internal or are there aspects of external moderation? What does it mean that “Examiners are provided protected time just before exams?” what are the examiners briefed on?

Our response: 

The pediatric exams are prepared by the staff in the department. Each member contributes in each exam. There is a blue print for each exam that it is followed. A departmental assessment committee, including academic staff in the department, evaluate each exam thoroughly. Then the exam is sent to the Faculty of Medicine exam committee for further evaluation and approval. Exam moderation is done internally. External examiners are involved as examiners in the final OSCE, in addition, they evaluate the written final exam (line 91, page 5).

“Examiners are provided protected time just before exams” Our response: before the start of the final OSCE, Examiners are assigned time to check on patients physical signs in the different exam stations and agree on what students should demonstrate (lines 96-98, page 5). 

what are the examiners briefed on? Our response: Few days prior to the final exam, the department of pediatrics, representing by the head of the department and the chair of the assessment committee and other members, hold an orientation session to the external examiners to explain the guidelines and the lay out of the exam, what standards are expected from the students so to define the pass/fail marks. The grading sheets and the global assessment sheets are discussed. (lines 105-108, page 6).

3. In the data collection, make it clear if you followed the same cohort of students over three years, which is what I think you are reporting, but it's not clear. You say that you enroll 100 students a year but do not explain why you do not have 300 students in your study. Who have you excluded?

Our response: We did not follow the same cohort for 3 years. We included the students rotated in pediatrics in the following academic years 2013-14, 2014-15 and 2015-16.

 The total number of students involved in the study was 286 students because the faculty admits, on average, 100 students every year, yet 14 students had early drop-out for different reasons (lines 67-69, page 4).

4. In the results section, you say that students who did their rotation last performed better than those who did it second, but not better that those who did it first. How do you explain the difference? Could this be attributed to the differences in the assessments?

Our response: There are no differences in assessments. We expected that students’ performance in the final OSCE is best for those who did their pediatrics rotation towards the end of the year and closer to the final exam. However, this was not consistent, especially when compared with those who did their rotation in the beginning of the academic year. Students, doing their pediatric rotation second (ie in the middle of the academic year), perform less in the final OSCE. An explanation to this would be the fact, the rotation in the middle of the year is interrupted by many holidays (New year, Christmas and our National days). The students might be distracted. Of note, this difference was only statistically significant when we compared the whole study population, it was not demonstrated among students in each year, or when compared last to beginning or middle to beginning of the year. (lines 225-228 p 12)

5. In the discussion, the 7th paragraph that talks to differences in performance based on the timing of the rotation, the inferences are not very clear and it's not clear how this is supported by the reported findings.

Our response: We reported no significant differences in the final OSCE performance between the 3 groups each year. The only significant difference was when we looked at the whole study sample, there was a slight improvement in the final OSCE performance between the students rotating last compared to those rotating in the middle of the year (Means= 79.17 vs 77.19, P=0.028) however, this was not true comparing the third rotation to the first, or second to first. Therefore, there might be no effect of the timing of the rotation on the final OSCE performance. Therefore, dividing students to do different rotations throughout the year, according to the capacity of the training centers, staff availabilities, applied curriculum, should not affect their performance in the final exams (lines 220-228, page12).

6. You conclude the paragraph by applauding the recall of information; in medical education, however, recall is not considered a very high value but rather information literacy and life-long learning capacity as the half-life of medical information is very short.

Our response: 

We agree. Recall of information was not meant. It is the life-long learning capacity and the ability to perform the skills learned fluently and efficiently it was clarified in line 232 page12.

Journal Requirements:

The format was modified accordingly

The manuscript was edited by AJE

 3. Thank you for stating in the manuscript Methods section:

'Ethical approval was obtained from The Health Sciences Ethical Committee. The use of students’ grades was approved by the Vice-Dean Academic Office at the Faculty of Medicine. Confidentiality of identity of students included was secured.' 

Ethical approval was obtained from the Health Sciences Center Ethical Committee, Faculty of Medicine, Kuwait University. The Board members are the Vice Dean of Research at the Faculty of Medicine as the head of the committee and five other members who are full professors in different specialties.

4. PLOS requires an ORCID iD for the corresponding author in Editorial Manager on papers submitted after December 6th, 2016. 

The ORCID ID of the corresponding author was updated in the Editorial Manager System 

ORCID ID: 0000-0001-6148-1707.

5. We note that you have indicated that data from this study are available upon request. PLOS only allows data to be available upon request if there are legal or

ethical restrictions on sharing data publicly. For information on unacceptable data access restrictions, please see http://journals.plos.org/plosone/s/data-availability#loc-unacceptable-data-access-restrictions.

The data will be available and shared. To be able to share the data, we need to apply for a permission from the administration at the Faculty of Medicine, which is a lengthy procedure and because of the COVID-19 pandemic and the lockdown situation in the country, we will not be able to have the permission in timely manner. We will be able to share the data when the situation is back to normal in the country. 

 6. Please include your tables as part of your main manuscript and remove the individual files. Please note that supplementary tables should be uploaded as separate "supporting information" files.

The tables were included as part of the main manuscript file. The supplementary files were removed.

Response to reviewer #1 comments:

Thank you for your valuable comments. We have responded and made necessary changes a follows:

5. Review Comments to the Author

Reviewer #1: I read the manuscript with great interest, I suggest the following comments to make it publishable.

• There is need to address other factors that may affect the final OSCE like gender, day of the exam (the authors mentioned that they have around 30 students /group and there were 9 stations run in 2 cycles, this mean that the exam either repeated for three days or in three different place), studying these factors will add to the quality of paper.

Our response: We agree. Gender and the day of the examination, if it is conducted in more than one day, are important factors that might affect the exam performance. In our medical school, the end of block OSCE is conducted in one day in 2 hospitals. The final OSCE is run over 2 day in 4 hospitals per day (lines 94-96 in page 5). A study on the effect of students’ demographic information, including gender, age, type of housing, order among siblings and others is being carried out. 

• Was the in-training OSCE formative without marks, did you provide any feedback for student in the in-training OSCE?

Our response: The in-training OSCE is graded. Feedback is provided to the students who failed the exam station only. During their 3 months of clerkship, the students perform Mini-Cex, where immediate feedback is given (lines 110-112, page 6)

• The term error bar is mentioned in the method section but it is not shown in the figure.

 Our response: This was a mistake. There was an error bar, but we opt to remove it in the final version of the manuscript as it does not add much to data analysis. This sentence should have been removed from the text prior to submission. Thank you for pointing it out.

 (Lines 125-126, page 7) was removed.

• In result section (‘End of block OSCE’ and ‘End of block written’ showed some improvement, especially in the year 2014-15 compared to 2013-14 and 2015-16 (p=0.016 & p=0.031) respectively.) could you clarify this sentence more, what it means and refer to table number. The word improvement here is not perfect with the meaning. (improvement mean that student score improved in this test in comparison to previous test)

Our response: The word improvement is misleading, I agree, as it implies comparing the performance of the same students over time. In this paragraph, we refer to table 1, showing the stability of our exam grades over the years, which make the study data comparable and hence, conclusions can be withdrawn. There are no significant differences between the grades of different exam components over the years. However, we think that it is confusing, so it was removed from the text ( lines 144-146, page 8).

• In discussion section, the authors should shed light on the low r value between written and final OSCE, usually the OSCE should be a test of clinical competence and not knowledge try to discuss this point when comparing final OSCE and written test, could be the good correlation between written test and OSCE due to more knowledge concentrated OSCE stations rather than performance OSCE stations.

Our response: We agree, OSCE should measure clinical competence. Many studies reported correlations between written exam performance and OSCE (3,8,18-20 in the references). As we have real patients in the OSCE run in our institution, we think that students should have a good knowledge base in order to perform well. Clinical findings should be sought as the student progresses with the patient encounter. They need to give explanations and answer few questions at the end of the station. The OSCE is conducted to ensure mastering the clinical skills. More clarification to the paragraph, with new references, was added (lines 203-219, pages 11,12).

• To mention reliability of a test there is need for reliability index.

Our response: True. The sentence was removed from the text as it might be confusing (line 205, page 11).

• In conclusion part the first sentence should be removed as it was not detected in the results.

Our response: Agree. The sentence was removed (lines 250,251, Page 13).

• Try to use some new references that addressed similar problem.

Alkhateeb, N. E., Al-Dabbagh, A., Ibrahim, M., & Al-Tawil, N. G. (2019). Effect of a Formative Objective Structured Clinical Examination on the Clinical Performance of Undergraduate Medical Students in a Summative Examination: A Randomized Controlled Trial. Indian pediatrics, 56(9), 745–748.

Our response: Thank you for the reference. This article and other 2 new references were added. References 14, 19 and 20.

English editing is advisable.

 Our response: The manuscript was edited by AJE

---

## [Editor Report · Decision Letter 1]

9 Jul 2020

Factors Predicting Students’ Performance in the Final Pediatrics OSCE

PONE-D-19-27962R1

Dear Dr.Maysoun Al Rushood ,

We’re pleased to inform you that your manuscript has been judged scientifically suitable for publication and will be formally accepted for publication once it meets all outstanding technical requirements.

Kind regards,

Oathokwa Nkomazana, MD MSC PhD

Academic Editor

PLOS ONE
---

## [Editor Report · Acceptance letter]

21 Aug 2020

PONE-D-19-27962R1 

Factors Predicting Students’ Performance in the Final Pediatrics OSCE 

Dear Dr. Al Rushood:

I'm pleased to inform you that your manuscript has been deemed suitable for publication in PLOS ONE. Congratulations! Your manuscript is now with our production department. 

Kind regards, 

on behalf of

Dr. Oathokwa Nkomazana 

Academic Editor

PLOS ONE